# Preparation of Nanoparticle Porous-Structured BiVO$_4$ Photoanodes by a New Two-Step Electrochemical Deposition Method for Water Splitting

**SocMan Ho-Kimura** [1,*]**, Wasusate Soontornchaiyakul** [2]**, Yuichi Yamaguchi** [2] **and Akihiko Kudo** [2]

[1] Institute of Applied Physics and Materials Engineering, University of Macau, Taipa, Macau SAR, China
[2] Department of Applied Chemistry, Faculty of Science, Tokyo University of Science, Tokyo 162-8601, Japan; wasu_soon@rs.tus.ac.jp (W.S.); y-yama@rs.tus.ac.jp (Y.Y.); a-kudo@rs.tus.ac.jp (A.K.)
[*] Correspondence: socmanho@um.edu.mo

**Abstract:** In the synthesis method of a BiVO$_4$ photoanode via BiOI flakes, a BiOI film is formed by electrochemical deposition in Step 1, and a vanadium (V) source solution is placed by drop-casting on the BiOI film in Step 2. Following this, BiVO$_4$ particles are converted from the BiOI–(V species) precursors by annealing. However, it is challenging to evenly distribute vanadium species among the BiOI flakes. As a result, the conversion reaction to form BiVO$_4$ does not proceed simultaneously and uniformly. To address this limitation, in Step 2, we developed a new electrochemical deposition method that allowed the even distribution of V$_2$O$_5$ among Bi–O–I flakes to enhance the conversion reaction uniformly. Furthermore, when lactic acid was added to the electrodeposition bath solution, BiVO$_4$ crystals with an increased (040) peak intensity of the X-ray diffractometer (XRD) pattern were obtained. The photocurrent of the BiVO$_4$ photoanode was 2.2 mA/cm$^2$ at 1.23 V vs. reversible hydrogen electrode (RHE) under solar simulated light of 100 mW/cm$^2$ illumination. The Faradaic efficiency of oxygen evolution was close to 100%. In addition, overall water splitting was performed using a Ru/SrTiO$_3$:Rh–BiVO$_4$ photocatalyst sheet prepared by the BiVO$_4$ synthesis method. The corresponding hydrogen and oxygen were produced in a 2:1 stoichiometric ratio under visible light irradiation.

**Keywords:** bismuth vanadate photoanode; electrochemical deposition; nanoparticle porous structure; water splitting; strontium titanate photocatalyst sheet

## 1. Introduction

Semiconductor photocatalysts and photoelectrodes can potentially be applied to a green technology that offers renewable energy alternatives to fossil fuels. One such technology is solar-driven hydrogen production by photoelectrochemical (PEC) water splitting. During PEC water splitting, hydrogen and oxygen are generated by p- and n-type semiconductor photoelectrodes, respectively. The advantage of the PEC system is that hydrogen production is separated from oxygen. Many n-type semiconductor photoelectrodes, especially the bismuth vanadate (BiVO$_4$) photoanode with favourable band edge positions, can also be utilised. The band gap of BiVO$_4$ is 2.4 eV and can absorb up to 11% of the solar spectrum assuming the theoretical photocurrent is 7.5 mA/cm$^2$ under AM1.5G one sun (100 mW/cm$^2$) illumination [1–3]. Its high flat band potential and being located close to the thermodynamic hydrogen evolution potential allow the water oxidation reaction to appear at a moderate external bias potential. Additionally, BiVO$_4$ can be easily synthesised and is relatively stable in aqueous solutions, non-toxic and inexpensive. The photocatalytic effect of BiVO$_4$ on water splitting under visible light was first reported by Kudo et al. [4]. Furthermore, Kudo et al. reported that scheelite monoclinic phase BiVO$_4$ showed higher visible light activity for water splitting than tetragonal phase BiVO$_4$ [5,6]. In 2003, Sayama et al. successfully demonstrated the PEC properties of a BiVO$_4$ thin film photoanode produced

by metal–organic decomposition [7]. Thereafter, numerous studies on $BiVO_4$-based thin film photoanodes have been reported. Van de Krol et al. investigated the behaviours of photo-charge carriers in $BiVO_4$ by microwave conductivity measurements, and the results reveal that $BiVO_4$ possesses poor carrier mobility of $4 \times 10^{-2}$ cm$^2$/Vs, a carrier lifetime of 40 ns and a carrier diffusion length of 70 nm [8]. To overcome these shortcomings, morphological control is one of the measures to improve the PEC properties efficiency of $BiVO_4$. A high photocurrent was obtained when $BiVO_4$ was placed on $WO_3$ nano-helixes [9], $WO_3$ nanowires [10], ZnO nanowires [11] or honeycomb template with ZnO [12]. This is because the nanostructures created huge surface area where the reaction sites of water oxidation increased. In addition, $WO_3$ and ZnO could work as a hole blocking layer, and the in-situ rectifying electron transferred across the heterojunction.

In 2012, Choi et al. developed a two-step method for synthesising $BiVO_4$ thin film with large surface-area and porous-structured [13]. Then, improved methods were announced in 2014 [14,15]. First, in the electrochemical deposition of $BiVO_4$, a metal bismuth film [14] or bismuth oxyiodide (BiOI) [13,15] film is formed on a fluorine-doped tin oxide (FTO) glass substrate by an electrochemical procedure. Then, these films are converted to $BiVO_4$ film using a vanadium source, such as vanadium pentoxide ($V_2O_5$), vanadyl acetylacetonate ($VO(acac)_2$), etc. When employing the method via BiOI [13,15,16], two-dimensional (2D) BiOI flakes are formed on an FTO substrate by electrochemical deposition. The BiOI is then converted to $BiVO_4$ worm-like particles with vanadium (V) species at high temperatures. However, the BiOI film surface is comparatively hydrophobic. Therefore, the uniform distribution of the V species in the BiOI film is challenging. Consequently, the reaction between the BiOI flakes and the V species proceeds randomly and nonuniformly within the BiOI thin film [17]. It is assumed that drop-casting of an organic solvent (e.g., dimethyl sulphoxide (DMSO)) containing $VO(acac)_2$ also cannot certainly penetrate among the 2D BiOI flakes uniformly. This may affect the uniformity of particle composition and particle size. An improvement in this study is to strengthen the dispersibility of V species all around the Bi species film and the balance of the Bi–V–O conversion reaction.

A combination of p- and n-type semiconductor photoelectrodes has been proposed for a PEC system with tandem cell for non-bias assisted water splitting. The tandem cell should be ideally irradiated from the semiconductor photoelectrode with a relatively wide band gap. Consequently, short wavelength lights are absorbed while the transmitted long wavelength lights are absorbed by another photoelectrode. To maximise the use of incident light on the electrode, radiation from the sample side (front illumination) is ideal. $BiVO_4$ is often used as the n-type wide band gap semiconductor for the PEC tandem cell system. There is a need for a $BiVO_4$ photoanode that has high incident light transmittance and excellent photoresponse from front illumination.

Large area photoelectrodes are also required for practical utilisation. However, increasing the area of a photoelectrode generally reduces its PEC performance [18,19]. A few reports revealed that the photocurrent did not proportionally increase with the area of the electrode. Pike et al. reported that photoelectrodes with areas of 25 and 300 cm$^2$ exhibited high initial photocurrents of 19.8 and 66.8 mA at 1.23 V vs. RHE, although they corresponded to low photocurrent densities of 0.82 and 0.22 mA/cm$^2$, respectively [18]. The lowering of the photocurrent density of the large photoanode is mainly attributed to resistances of the FTO substrate [18,19]. The substrate and electrolyte conductivities, electrode configuration uniformity, electrode pH stability, etc. dominate overall PEC performance of water splitting [19]. It has been reported that carrier transfer in an FTO substrate was improved by coating metal lines or p-type semiconductor lines on the device [19,20]. Despite this benefit, to our knowledge, the utilisation of FTO substrate photoanodes with an area bigger than 1 cm$^2$ has seldom been reported for PEC water splitting.

Conversely, it is well known that the photocurrent of $BiVO_4$ is majorly limited by surface recombination [21]. Loading a passivation cocatalyst on the surface, such as cobalt phosphate complex [22], cobalt oxide [23], iron oxyhydroxide [13,24], NiFe–(oxy)hydroxide [16] and a double layer of FeOOH–NiOOH [15], could enhance the transfer

of charge carriers at the interface between a photoanode and an electrolyte. The mechanism of the effect of cobalt oxide on $BiVO_4$ is the same as that on hematite. Divalent cobalt can be oxidised into trivalent and tetravalent cobalt, which can be reduced back to divalent cobalt, and this catalytically assists water oxidation [22].

In this study, we synthesised a highly functional porous $BiVO_4$ photoelectrode by a new two-step electrochemical deposition method. Our strategy was to enhance the dispersibility of V species throughout the Bi species film and the balance of the Bi–V–O conversion reaction to form a uniform $BiVO_4$ film. To achieve this, we focused on the electrodeposition bath solution of the BiOI in Step 1. Moreover, in Step 2, amorphous $V_2O_5$ was formed among Bi–O–I flakes by an electrochemical deposition method rather than a drop-casting method. Such electrochemical deposition method is attractive because electrochemical deposition is widely adopted in industries. The synthesis of $BiVO_4$ by electrochemical deposition offers easy scalability at a comparatively low cost. Here, we discuss the formation mechanism, morphology and characterisation of the $BiVO_4$ thin film photoelectrodes. Further, the benefit of film area up to $\geq 3 \text{ cm}^2$ was demonstrated to describe the shortcomings of large area photoelectrodes. The overall water splitting was also confirmed using a $Ru/SrTiO_3$:Rh–$BiVO_4$ photocatalyst sheet prepared by the present $BiVO_4$ synthesis method as an application example.

## 2. Results and Discussion

### 2.1. Film Preparation

#### 2.1.1. Formation of $BiVO_4$

Choi et al. developed the two-step $BiVO_4$ synthesis method, which gave an excellent performance for PEC water oxidation [13,15]. In the Choi method, in Step 1, BiOI was formed by electrochemical deposition. In Step 2, the V source (e.g., $V_2O_5$, VO(acac)$_2$, etc.) solution was placed by drop-casting on the BiOI film. Following this, the Bi–V–O reaction was processed by heating at high temperatures (350–650 °C) for 2–5 h in air to convert the BiOI–V precursor into $BiVO_4$. In our tests, when using the method of placing a solution of vanadium species on the BiOI film by drop-casting, the $BiVO_4$ thin film obtained after the Bi–V–O heat treatment often broke. There is still room for improvement in the $BiVO_4$ crystal growth process for PEC properties. First, we focused on the components of the electrodeposition bath solution. It is well known that, in the field of electrochemical deposition research, lactic acid ($C_3H_6O_3$) in the deposition bath forms complexes with transition metals (e.g., Cu) and affects the crystal growth in a particular orientation [25]. Figure 1 shows the flowchart for preparing our $BiVO_4$ photoanode; the procedure in Step 1 is similar to the Choi method [13,15]. As a standard for this study, the aqueous deposition bath solution in Step 1 contained 60 mmol/L lactic acid. In Step 1, the complexes of Bi with iodide $[BiI_4]^-$ and lactic acid $[Bi(C_3H_4O_3)_2(C_3H_5O_3)]^{2-}$ were first produced at pH 1.2 in the aqueous solution bath (Equations (1) and (2)). Then, the solution of the $Bi^{3+}$ species was mixed with an ethanol benzoquinone solution. The bath solution was stirred overnight for complex formation and pH stabilisation. Some of the benzoquinones were reduced to hydroquinone, thereby shifting the pH value towards alkalinity (pH 3.6–4.0). During Step 1, cathodic deposition, benzoquinone was actively reduced to hydroquinone (Equation (3)). At the same time, the concentration of the hydroxide ions ($OH^-$) increased on the working electrode and reacted with the bismuth iodide $[BiI_4]^-$ species to form BiOI (Equation (4)). Next, in our method, to reduce the hydrophobicity of the film surface, the BiOI film was heated to 450 °C for 1 h to generate penta-bismuth hepta-oxide iodide ($Bi_5O_7I$). In Step 2, an ethanol aqueous solution of vanadyl sulphate ($VOSO_4$) was utilised as the electrodeposition bath, and $VO^{2+}$ was oxidised to amorphous $V_2O_5$ (Equation (5)) throughout the $Bi_5O_7I$ film during the anodic deposition. This process allowed an aqueous solution of vanadium source to penetrate among the $Bi_5O_7I$ flakes to produce $V_2O_5$. This process is different from previous reports. The as-deposited films were annealed at 475 °C for 1 h in air to convert the $Bi_5O_7I$–$V_2O_5$ precursor to $BiVO_4$. The pure $BiVO_4$ was obtained after washing with an aqueous KOH solution (0.5 mol/L) to remove residual $V_2O_5$. It

seems that the adhesion between the FTO substrate and the BiVO$_4$ film was strong. The BiVO$_4$ film prepared by the electrodeposition method could not be completely removed from the FTO substrate by wiping it strongly with tissue paper.

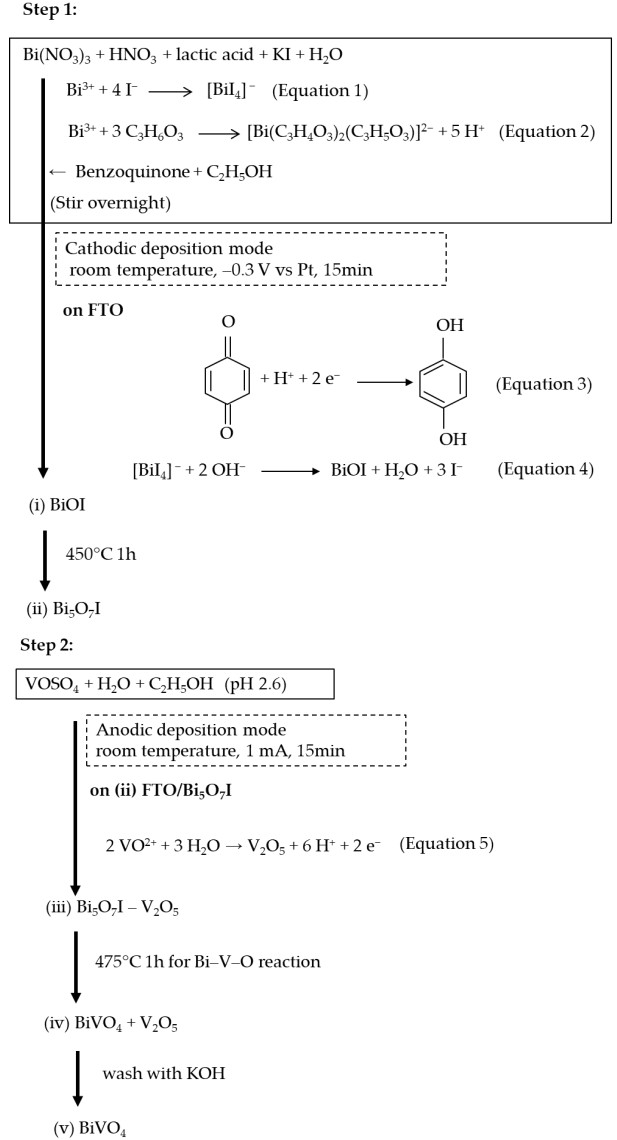

**Figure 1.** Flowchart of the new two-step electrochemical deposition method for preparing the BiVO$_4$ photoanode.

$$Bi^{3+} + 4I^- \rightarrow [BiI]^- \tag{1}$$

$$Bi^{3+} + 3C_3H_6O_3 \rightarrow [Bi(C_3H_4O_3)_2(C_3H_5O_3)]^{2-} + 5H^+ \tag{2}$$

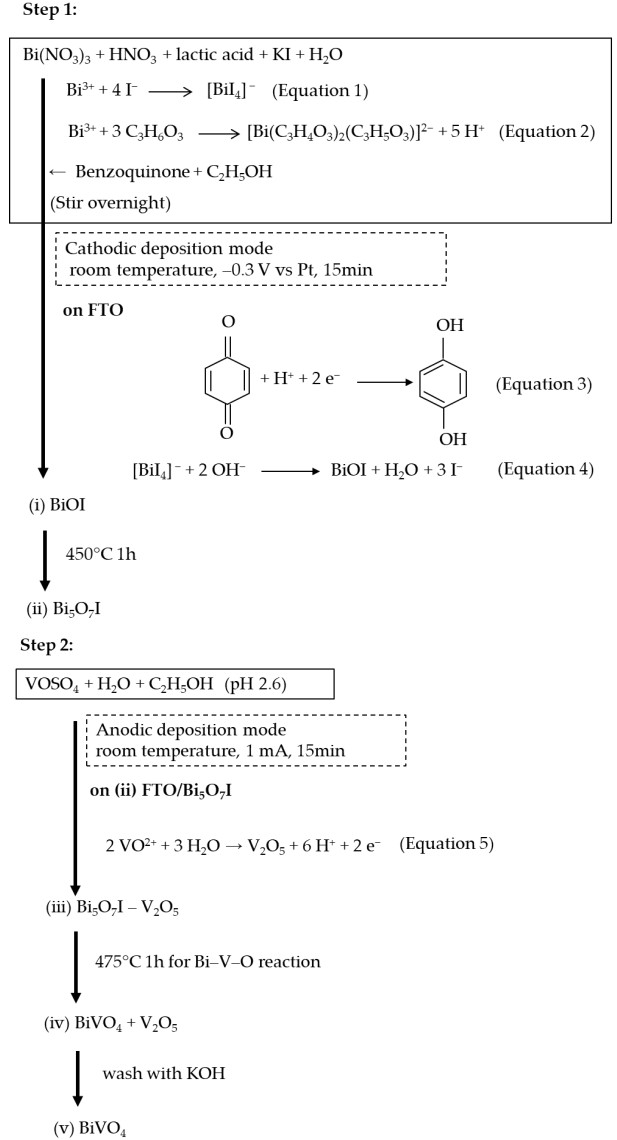

$$\tag{3}$$

$$[BiI_4]^- + 2OH^- \rightarrow BiOI + H_2O + 3I^- \tag{4}$$

$$2VO^{2+} + 3H_2O \rightarrow V_2O_5 + 6H^+ + 2e^- \tag{5}$$

BiOI, $Bi_5O_7I$, and $BiVO_4$ were synthesised according to the procedure shown in the flowchart in Figure 1. The absorbances of the BiOI, $Bi_5O_7I$, and $BiVO_4$ films agreed with those in the previous literature [15,26] (Figure S1). BiOI and $Bi_5O_7I$ produced by Step 1 were confirmed by X-ray diffraction (XRD) (Figure 2A). In the new synthesis Step 2 (Figure 2B), the as-deposited $V_2O_5$ was amorphous and could not be detected by XRD. However, after mild heating for 1 h at 475 °C, the XRD pattern of a mixed pattern of $BiVO_4$ and $V_2O_5$ was observed. Thereafter, the excess $V_2O_5$ was removed with an aqueous KOH solution. In Figure 2 (Bv), the paired peaks, which are characteristic of the scheelite-structured monoclinic $BiVO_4$, appeared around 19° and 35° of 2θ.

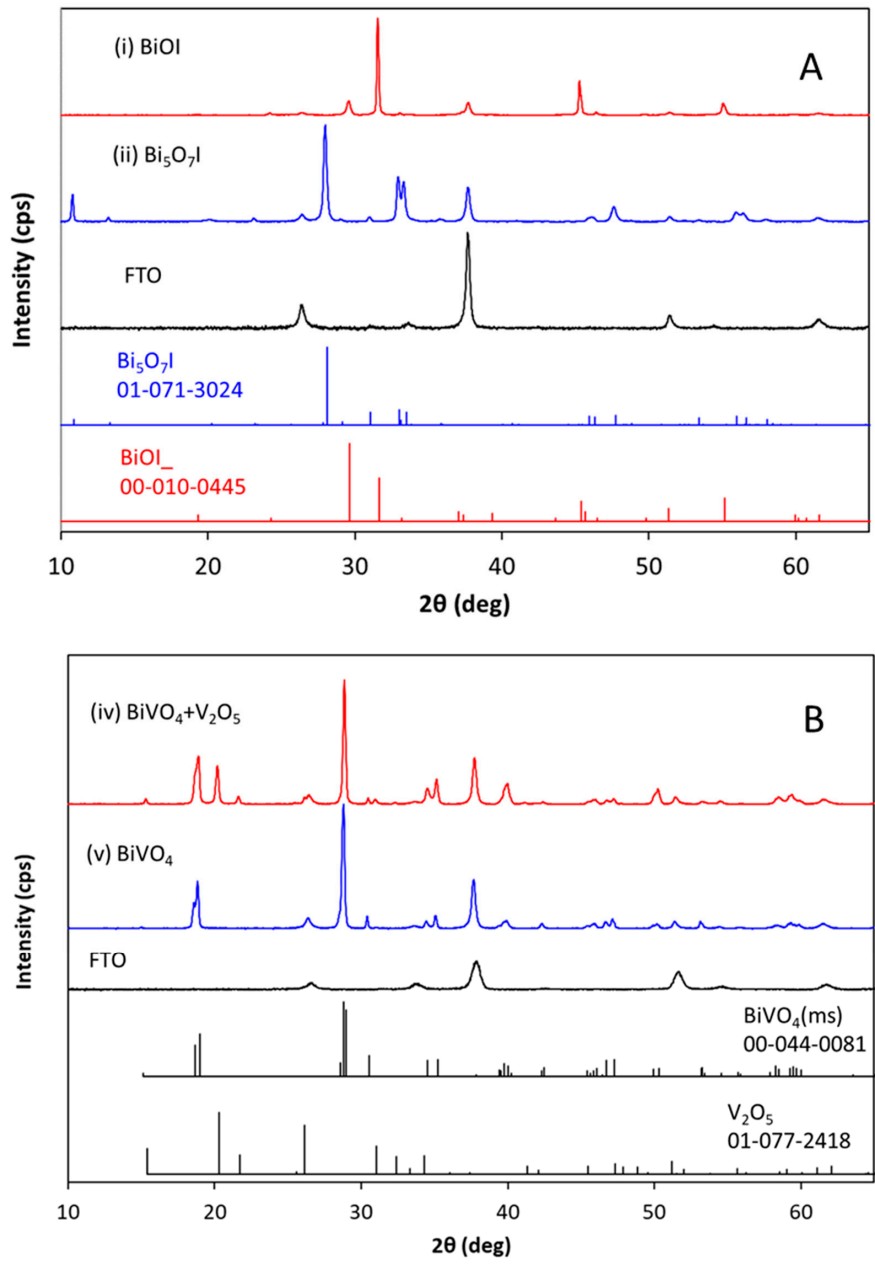

**Figure 2.** XRD patterns of: (**A**) Synthesis procedure Step 1 ((**i**) BiOI; and (**ii**) $Bi_5O_7I$ film and FTO substrate). For reference, $Bi_5O_7I$ (JCPDS No. 01-071-3024) and BiOI (JCPDS No. 00-010-0445) are cited. (**B**) Synthesis procedure Step 2 ((**iv**) $BiVO_4$ + $V_2O_5$ film; and (**v**) pure $BiVO_4$ film and FTO substrate). For reference, monoclinic scheelite $BiVO_4$ (JCPDS No. 00-044-0081) and $V_2O_5$ (JCPDS No. 01-077-2418) are cited. The numbers (**i,ii,iv,v**) correspond to the samples in Figure 1.

Regarding morphological observations, the BiOI film was relatively hydrophobic because of the physical structure of the 2D BiOI flake (Figure 3i and Figure S2i). After the heat treatment at 450 °C for 1 h, the BiOI turned to $Bi_5I_7O$. The sharp BiOI flakes became the rounded tip of $Bi_5O_7I$, which caused a difference in surface height, and succeeded in reducing the hydrophobicity of the film surface (Figure 3ii and Figure S2ii). The cross-sectional image of the $Bi_5O_7I$–$V_2O_5$ film (Figure 3iii) displays the $Bi_5O_7I$ flakes surrounded by the as-deposited amorphous $V_2O_5$. The thickness of the $Bi_5O_7I$–$V_2O_5$ thin film was approximately 350 nm.

The atomic distribution of Bi and V was performed by energy-dispersive X-ray spectroscopy (EDS) mapping to investigate whether the Bi–V–O conversion reaction could proceed throughout the thin film at the same time. In Figure 4, evidently, the EDS elemental mapping images of the $Bi_5O_7I$–$V_2O_5$ film shows V atoms were densely and uniformly dispersed all over the $Bi_5O_7I$ film. The atomic ratio of Bi to V was 1:2.17, which contained extra $V_2O_5$ for $BiVO_4$ conversion.

During the annealing process at 475 °C for 1 h, $BiVO_4$ particles were formed by reacting the $V_2O_5$ and the $Bi_5O_7I$ flake on the FTO substrate. Since the Bi and V atoms were uniformly present, it is considered that the Bi–V–O conversion reaction occurred throughout the Bi species thin film and at the same time. Consequently, a porous structured thin film was created with nanosized $BiVO_4$ particles. In Figure 5, the shape of the $BiVO_4$ nanoparticles was not the same as the worm-like particles prepared by the Choi method [13,15,16]. These images indicate that the $BiVO_4$ nanoparticles were highly monodispersed. If extra agglomerates of $BiVO_4$ particles were formed on the surface by the excessive progress of Bi–V–O reaction during heating process, the agglomerates of $BiVO_4$ particles can act as carrier traps that might reduce the photocurrent [16]. However, Figure 5a shows that no unwanted $BiVO_4$ particle agglomerates on the surface of the $BiVO_4$ thin film. Furthermore, the EDS composition analysis of the pure $BiVO_4$ film resulted that the rational atomic ratio of Bi to V was 1:1.08 (Figure S3).

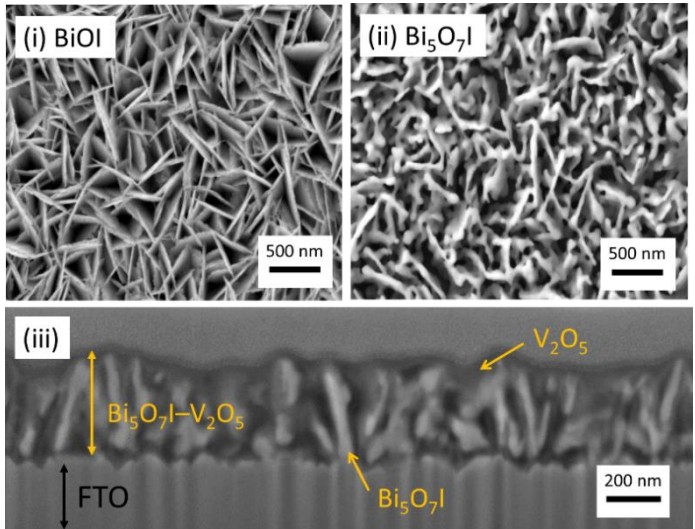

**Figure 3.** Top-view scanning electron microscope (SEM) images of: (**i**) BiOI; and (**ii**) $Bi_5O_7I$. The focused ion beam SEM (FIB-SEM) cross-sectional image of (**iii**) the as-deposited amorphous $V_2O_5$ distributed in the $Bi_5O_7I$ film.

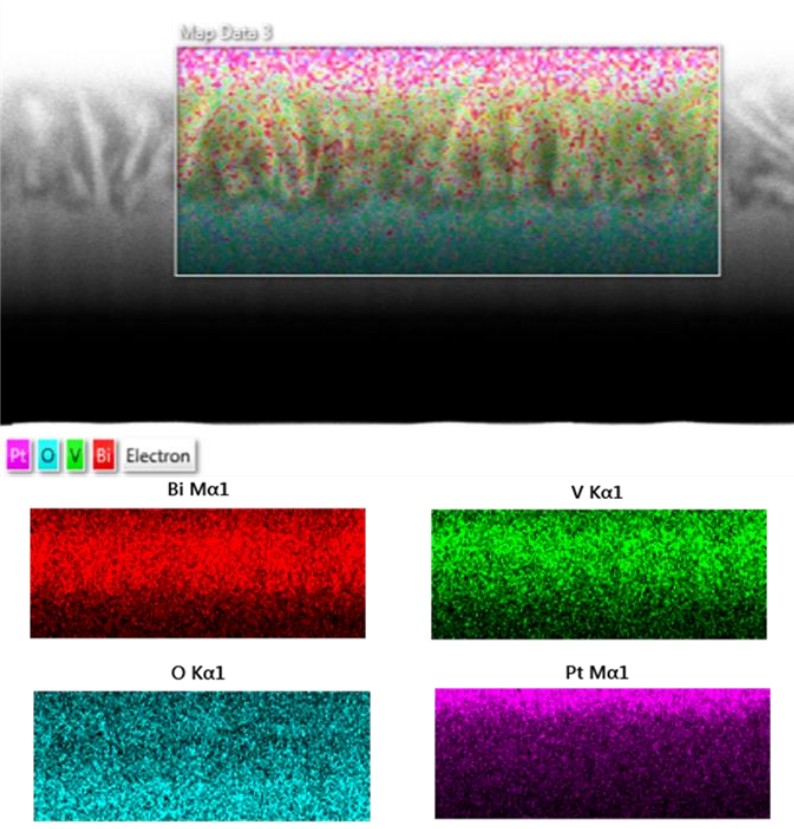

**Figure 4.** Cross section mapping images of the as-deposited amorphous $V_2O_5$ distributed in the $Bi_5O_7I$ film (in Figure 1iii) by EDS elemental analysis using an FIB-SEM. Platinum (Pt) is the conductive coating for the FIB-SEM measurement.

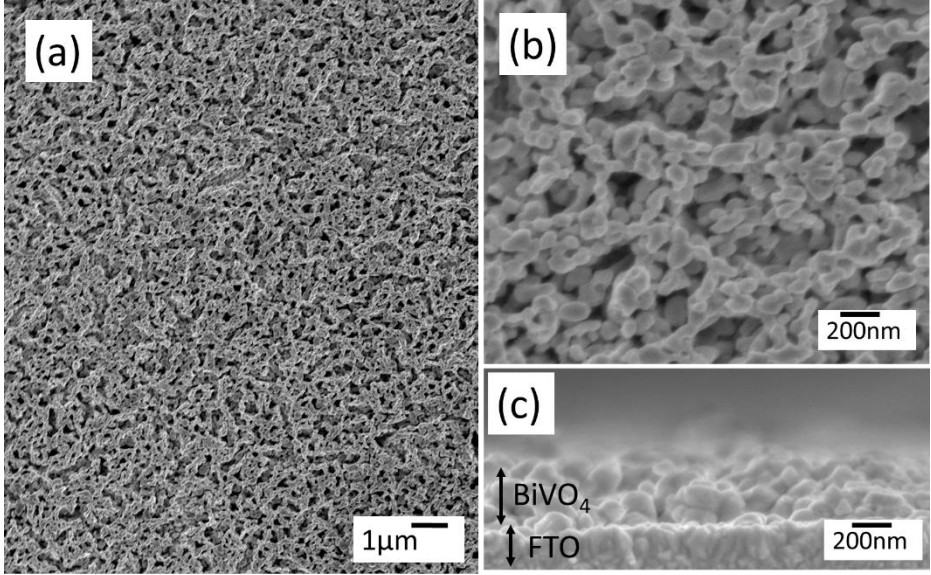

**Figure 5.** SEM images of the morphology-controlled $BiVO_4$ thin film in Figure 1v. Top views of: (**a**) low magnification; (**b**) high magnification; and (**c**) cross section.

### 2.1.2. Effect of Adding Lactic Acid

To investigate the effect of the addition amount, $BiVO_4$ films were prepared by changing the addition amount of lactic acid according to the procedure shown in Figure 1. The XRD patterns of the $BiVO_4$ films all showed monoclinic scheelite-structure with a (040)

peak of about 31° at 2θ (Figure 6A). In particular, the XRD pattern of $BiVO_4$ formed using the electrodeposition bath contained 60 mmol/L lactic acid exhibited increased (040) peak intensity. The (040) peak of the XRD pattern corresponds to the (010) facet. It is presumed that BiOI produced in the presence of an appropriate amount of lactic acid strongly adhered to the FTO in the specific orientation, hence the $BiVO_4$ (010) facets exposed favourably after $Bi_5O_7I$ reacted with $V_2O_5$ at 475 °C.

The $BiVO_4$ particles converted by the BiOI–(V species) precursors [15] on an indium tin oxide (ITO) substrate held a comparatively low light transmittance of ca. 30% at 600 nm [16]. This transmittance percentage was equivalent to the case without lactic acid in this study. As shown in Figure 6B, the $BiVO_4$ thin film prepared using the electrodeposition bath containing 60 mmol/L lactic acid shows the highest transmittance spectrum and has a transmittance of 52% at 600 nm. It is considered that the transmittance was improved because of atomic arrangement of the (010) oriented $BiVO_4$. From the perspective of transmittance, our $BiVO_4$ photoanode will also be suitable for practical application of a PEC tandem cell system.

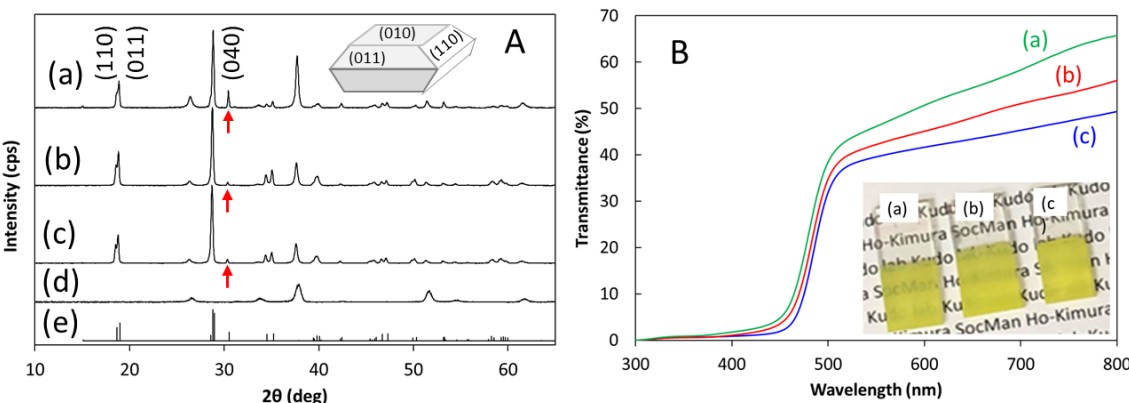

**Figure 6.** (**A**) The XRD patterns of $BiVO_4$ films prepared according to Figure 1. The electrodeposition bath used in Step 1 contained: (**a**) 60 mmol/L lactic acid; (**b**) 30 mmol/L lactic acid; and (**c**) lactic acid-free. The XRD patterns of: (**d**) FTO substrate; and (**e**) monoclinic scheelite-structured $BiVO_4$ (JCPDS No. 00-044-0081). The arrow points to the (040) peak that corresponds to the (010) facet. The illustration shows the $BiVO_4$ crystal facets. (**B**) Transmittance spectra of $BiVO_4$ films in (**A**). Transmittance spectra were measured by a spectrophotometer with an integrating sphere that was operated in the transmission mode. Inset is a photo of these $BiVO_4$ films.

### 2.1.3. Conversion Temperature from $Bi_5O_7I$–$V_2O_5$ Precursor to $BiVO_4$

The effect of temperature on the conversion of $Bi_5O_7I$–$V_2O_5$ precursors of Sample (iii) in Figure 1 to $BiVO_4$ crystals is especially important. The annealing process of the Bi–V–O conversion reaction was investigated from 350 to 550 °C. In the XRD patterns (Figure 7 and Figure S4), all the thin films exhibited the (040) peak after heat process at 350–550 °C. For the paired peak, a feature of the scheelite-structured monoclinic $BiVO_4$, the required temperature for the Bi–V–O annealing process was 450 °C or higher. A tetragonal $BiVO_4$ was established by heating at 400 °C or lower. The $BiVO_4$ thin film prepared at 350 °C exhibited relatively low crystallinity.

In the SEM images (Figure S5), it was observed that the particle size of $BiVO_4$ increased as the temperature was increased. These particles exhibited high uniformity in the size and shape of each film. Furthermore, there was no extra agglutination of $BiVO_4$ particles on the surface of these films. At the process temperature from 350 to 475 °C, the nanoscale particles created porous structured layer on the FTO substrate. Noticeably, these thin films possessed a relatively large surface area. Contrarily, the aggregation of the $BiVO_4$ particles was initiated at 500 °C, and the porous structure became non-porous. At 550 °C, completely large particles were remodelled into a non-porous-layered structure. Therefore, when the conversion temperature was extremely high, the surface area where the water oxidation

reaction occurs decreased. To prevent the agglutination of ultrafine particles, it is vital to complete the conversion reaction in a short time such as 1 h. Additionally, the light scattering between the large particles occurred in the 500 and 550 °C treated thin films, and the absorbance increased in the long-wavelength region (Figure S6). The samples treated at 450 and 475 °C possessed the appropriate band gaps for $BiVO_4$ and enabled the absorption of visible light of up to 520 nm.

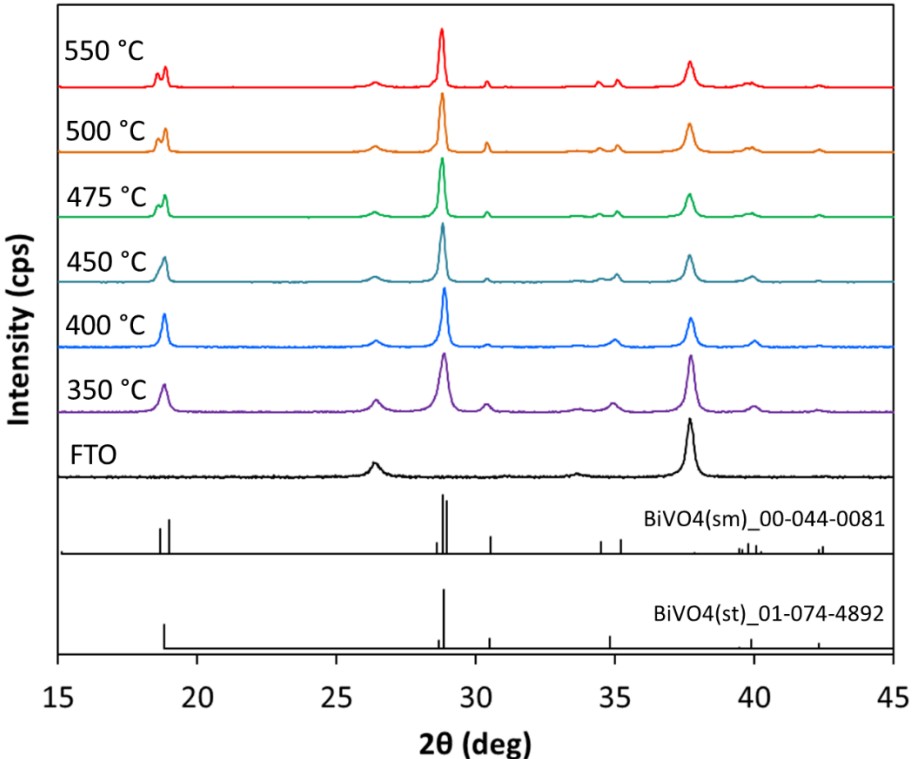

**Figure 7.** XRD patterns of $BiVO_4$ films grown on FTO substrate by $Bi_5O_7I$ and $V_2O_5$ reaction treated at several different temperatures, from the top to the bottom: 550, 500, 475, 450, 400 and 350 °C, as well as the FTO substrate. For reference, scheelite-structured monoclinic $BiVO_4$ (JCPDS No. 00-044-0081) and scheelite-structured tetragonal $BiVO_4$ (JCPDS No. 01-074-4892) are cited.

### 2.2. PEC Properties

#### 2.2.1. Effect of Adding Lactic Acid

The photocurrent density–potential (J–V) curves of $BiVO_4$ films grown in the deposition bath with varying lactic acid concentrations were measured (Figure 8). From the photocurrent density, it was observed that the optimum concentration of lactic acid added to the deposition bath was 60 mmol/L. This result may be related to the XRD findings in Figure 6A. $BiVO_4$ crystals with exposed (010) facets corresponding to the (040) peak of the XRD pattern were reported to exhibit higher photocatalytic activity for water oxidation than without (040) peak [27]. However, even with high peak intensities, the improvement in water oxidation reaction did not increase proportionally [27].

On the other hand, the (010) and (110) crystal facets of monoclinic $BiVO_4$ have been reported to be active reducing and oxidising sites, respectively [28]. The large (010) facet provided rapid charge separation and suppressed both bulk and surface recombination [28]. In this study, the (010) oriented $BiVO_4$ crystals enhanced the photocurrent density by improving electron transport.

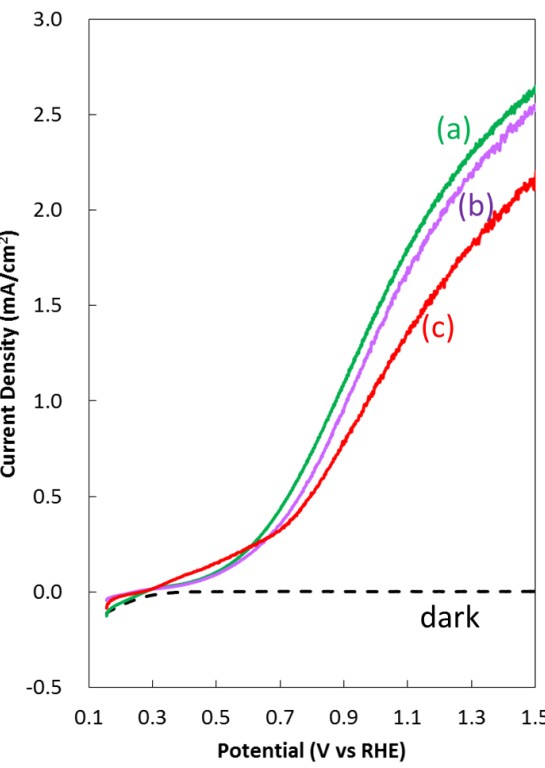

**Figure 8.** J–V curves of $BiVO_4$ thin films prepared according to Figure 1. The electrodeposition bath used in Step 1 contained: (**a**) 60 mmol/L lactic acid; (**b**) 30 mmol/L lactic acid; and (**c**) lactic acid-free. The dark current is shown as a dashed line. Data were collected in 0.5 mol/L potassium borate aqueous solution (pH 9.5), under solar simulated light of 100 mW/cm$^2$ illumination.

### 2.2.2. Bi–V–O to $BiVO_4$ Conversion Temperature

The temperature-dependent PEC properties of the Bi–V–O annealing process were investigated from 350 to 550 °C (Figure 9A). The $BiVO_4$ photoanodes processed by the Bi–V–O annealing at 450 and 475 °C showed excellent photocurrent densities. Figure 9B shows the applied bias photo-to-current efficiency (ABPE) calculated from the corresponding J–V curves. Generally, the average maximum ABPE of bare $BiVO_4$ photoanode is 0.3% or less in the range of about 0.9–1.0 V vs. RHE [29,30]. In the present study, for the Bi–V–O annealing process at 475 °C, the ABPE of the bare $BiVO_4$ photoanode reached 0.37% at 0.9 V vs. RHE, which was the greatest of our tests. This is probably because the porous-structured $BiVO_4$ photoanode was a highly crystalline scheelite-structured monoclinic system.

Furthermore, the open circuit (OC) potential of each photoanode was measured in the dark and under illumination (Figure S7A) because the difference in the OC potentials between dark and light conditions is related to the photovoltage. Further, the onset potential is considered to be related to the light OC potential. The $BiVO_4$ photoanode prepared by Bi–V–O annealing process at 475 °C recorded the most negative light OC potential and the biggest difference (Figure S7B). It can be inferred that the photoanode produced at 475 °C possessed a higher photovoltage than the other temperatures. These characteristics discussed above are consistent with those of the PEC photoresponse.

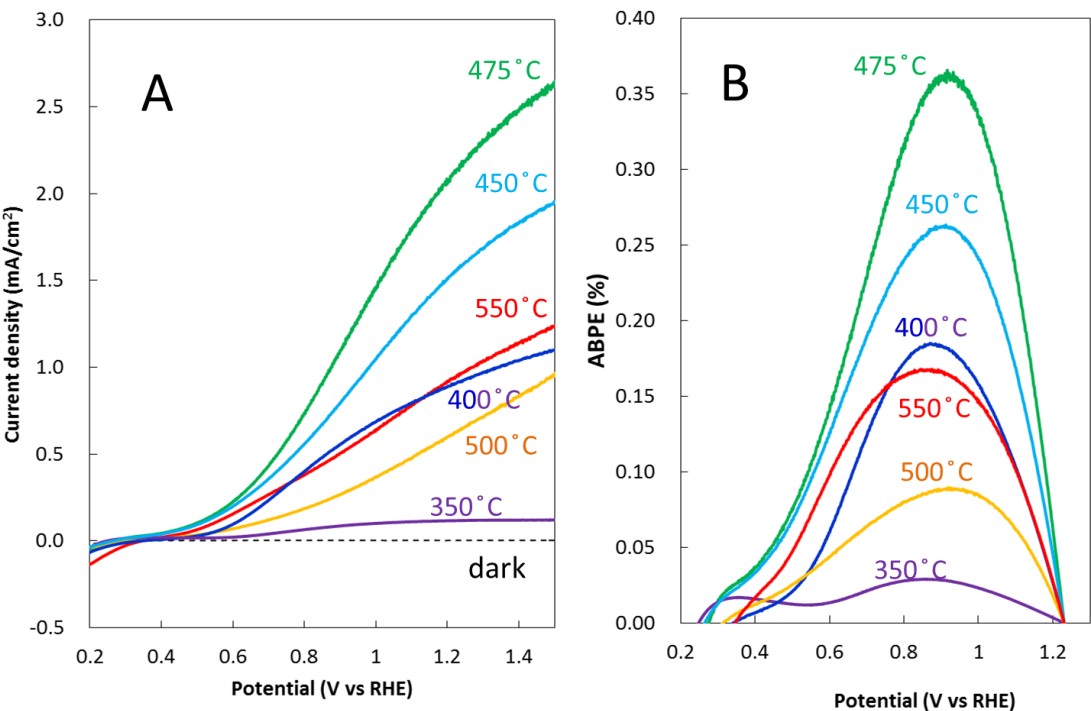

**Figure 9.** (**A**) J–V curve of BiVO$_4$ films made by Bi–V–O conversion process at varying temperature from 350 to 550 °C according to Figure 1. (**B**) Applied bias photo-to-current efficiency (ABPE) was calculated from the J–V curves in (**A**). Data were collected in 0.5 mol/L potassium borate aqueous solution (pH 9.5) under solar simulated light of 100 mW/cm$^2$ illumination.

### 2.2.3. PEC Properties of the Optimised BiVO$_4$ Photoanode

The optimised bare BiVO$_4$ photoanode was obtained by using the electrodeposition bath containing 60 mmol/L lactic acid and treating the Bi–V–O conversion process at 475 °C. The photocurrent density of water oxidation attained 2.2 mA/cm$^2$ at 1.23 V vs. RHE (Figure 10A). In addition, using an electrolyte with hole scavenger K$_2$SO$_3$, the J–V curve shifted to the negative potential region and the photocurrent density increased significantly. It is considered that the oxidation of K$_2$SO$_3$ reduced the charge recombination on the surface of BiVO$_4$.

To quantify the contributions of the suppressing bulk or surface recombination, charge separation efficiency ($\eta_{sep}$) and surface transfer efficiency ($\eta_{trans}$) of the BiVO$_4$ photoanode were calculated by the following equation: [22,31]

$$J_{H_2O} = J_{absorbed} \times \eta_{trans} \times \eta_{sep} \tag{6}$$

$$J_{K_2SO_3} = J_{absorbed} \times \eta_{sep} \tag{7}$$

$$\eta_{sep} = J_{K_2SO_3}/J_{absorbed} \tag{8}$$

$$\eta_{trans} = J_{H_2O}/J_{absorbed} \times \eta_{sep} = J_{H_2O}/J_{K2SO3} \tag{9}$$

where J$_{H_2O}$ is the measured photocurrent density and J$_{absorbed}$ is the photon absorption rate (expressed as a photocurrent density). The calculated J$_{absorbed}$ of the BiVO$_4$ photoanode was specified as 4.45 mA/cm$^2$ [15]. J$_{H2O}$ and J$_{K_2SO_3}$ are the photocurrent densities collected in the 0.5 mol/L potassium borate electrolyte without and with 0.2 mol/L K$_2$SO$_3$, respectively. When K$_2$SO$_3$ is added into an electrolyte, it acts as a hole scavenger. In the presence of K$_2$SO$_3$ in electrolyte, the surface transfer efficiency is assumed to be 100%.

For comparison, in general, the average charge separation efficiency ($\eta_{sep}$) of bare BiVO$_4$ photoanode was lower than 45% at 1.23 V vs. RHE [30,31], and the average sur-

face transfer efficiency ($\eta_{trans}$) was ca. 35% at 1.23 V vs. RHE [30,32,33]. As shown in Figure 10B, the $\eta_{sep}$ and $\eta_{trans}$ of the bare $BiVO_4$ increased to 56% and 86% at 1.23 V vs. RHE, respectively. This is likely due to the uniform particle size and composition of the morphologically controlled $BiVO_4$ crystals, which increased the efficiency of the water photooxidation reaction.

The exposed (010) facets are able to provide rapid electron transfer, thus improving the PEC water oxidation on $BiVO_4$ by smooth charge separation [27,28]. Our results indicate that lactic acid contributed to the growth of (010) oriented crystals. In addition, by controlling the Bi–V–O reaction temperature, particles of appropriate size were supplied. The hole diffusion length of $BiVO_4$ is estimated as 70 nm [8]; nanosized particles are more advantageous for the movement of holes to the surface. Consequently, the charge recombination was reduced, thereby resulting in an increased photocurrent density even with frontal illumination.

The photocurrent from the back illumination was slightly lower than that from the front illumination (Figure 10C). In many reports, large $BiVO_4$ crystals size and film thickness resulted in higher photocurrents at the back exposure than at the front [34,35]. In contrast, our $BiVO_4$ film was composed of nanosized crystals, was thin, and showed high transmittance. In the case of the back illumination, there was probably a loss of light by the reflection on the glass surface of the FTO substrate.

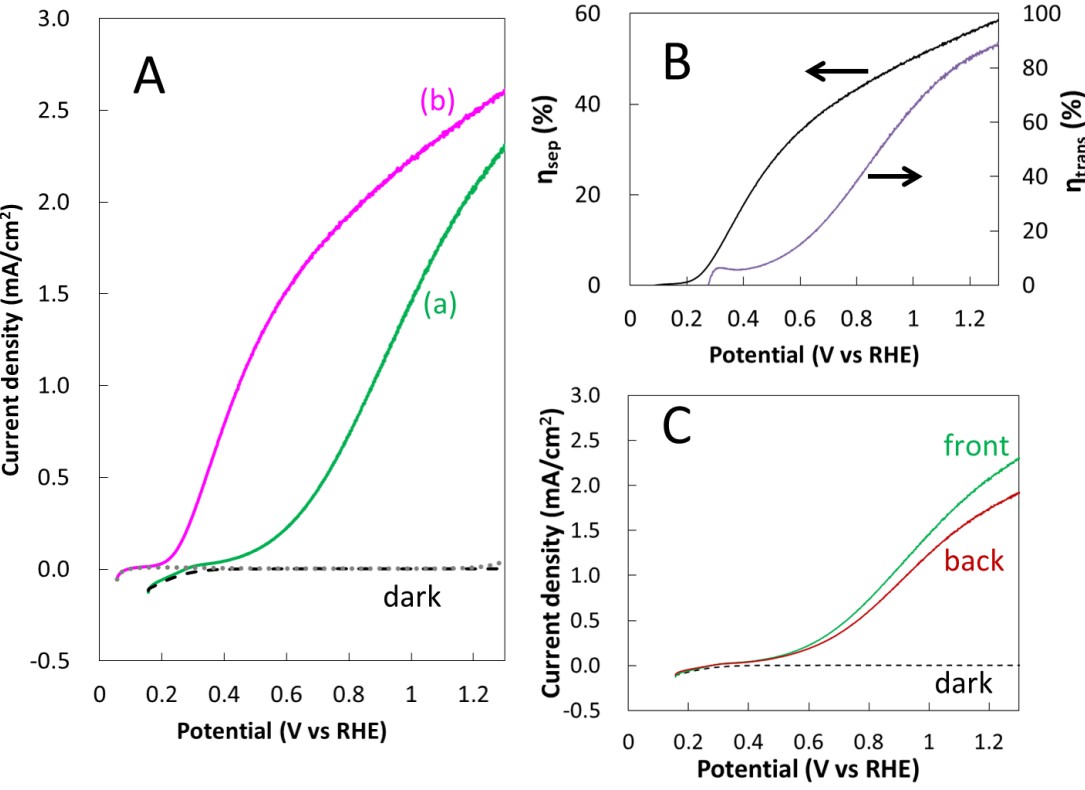

**Figure 10.** (**A**) J–V curves of the optimised bare $BiVO_4$ photoanode measured in 0.5 mol/L potassium borate (pH 9.5) (**a**) without and (**b**) with 0.2 mol/L $K_2SO_3$ as the hole scavenger; (**B**) $\eta_{sep}$ and $\eta_{trans}$ calculated from the J–V curves in (**A**); and (**C**) J–V curves of the bare $BiVO_4$ photoanode from the front and back (the FTO side) illuminations. The dark current is shown as a dashed line. Data were collected under solar simulated light of 100 mW/cm² illumination.

### 2.2.4. Effect of Cocatalyst

It is known that surface modification of $BiVO_4$ with a cocatalyst can reduce the recombination of the surface charges and accelerate the rate of oxygen generation [21,24]. It was confirmed that $BiVO_4$ utilised in this study was also influenced by the loading of

cobalt (II, III) oxide ($Co_3O_4$) cocatalyst (Figure 11A). The $BiVO_4/Co_3O_4$ photoelectrode was obtained by drop-casting an ethanol cobalt nitrate solution to the $BiVO_4$ photoanode surface and heating it to 350 °C for 1 h in air. The cobalt nitrate decomposed into $Co_3O_4$ during heating process at 350 °C [23].

The photocurrent of the bare $BiVO_4$ and $BiVO_4/Co_3O_4$ photoelectrodes were 2.2 and 2.9 mA/cm$^2$ at 1.23 V vs. RHE, respectively (Figure 11A). The $Co_3O_4$ cocatalyst exerted a great influence on the negative potentials shift of the J–V curve, which was a characteristic of the effect of cocatalysts. Moreover, when using the electrolyte with $K_2SO_3$ of a hole scavenger, a higher photocurrent of 3.8 mA/cm$^2$ at 1.23 V vs. RHE was achieved.

Owing to the effect of the $Co_3O_4$ cocatalyst, the maximum ABPE of water oxidation increased from 0.37% at 0.9 V to 1.03% at 0.6 V vs. RHE (Figure 11B). The $Co_3O_4$ cocatalyst effectively enhanced the kinetic water oxidation on the $BiVO_4$ photoanode, and a higher efficiency was obtained with moderate bias.

The incident photon-to-current conversion efficiency (IPCE) measurements were performed by monochromatic lights between 370 and 560 nm to ascertain the light conversion efficiency (Figure 12). Typically, the average IPCE value at 420 nm performed at 1.23 V vs. RHE is lower than 20% for bare $BiVO_4$ [11,35,36]. In the present study, the IPCEs of the bare $BiVO_4$ photoanode were 7.5% and 23.2% at 0.6 and 1.23 V vs. RHE, respectively, at 420 nm. Further, the IPCEs of the $BiVO_4/Co_3O_4$ photoanode exhibited remarkable increase to 23.8% and 37.5% at 0.6 and 1.23 V vs. RHE, respectively, at 420 nm. Both $BiVO_4$ photoanodes with and without $Co_3O_4$ cocatalyst presented photoresponse in the visible light region, which was consistent with the absorption spectrum of $BiVO_4$.

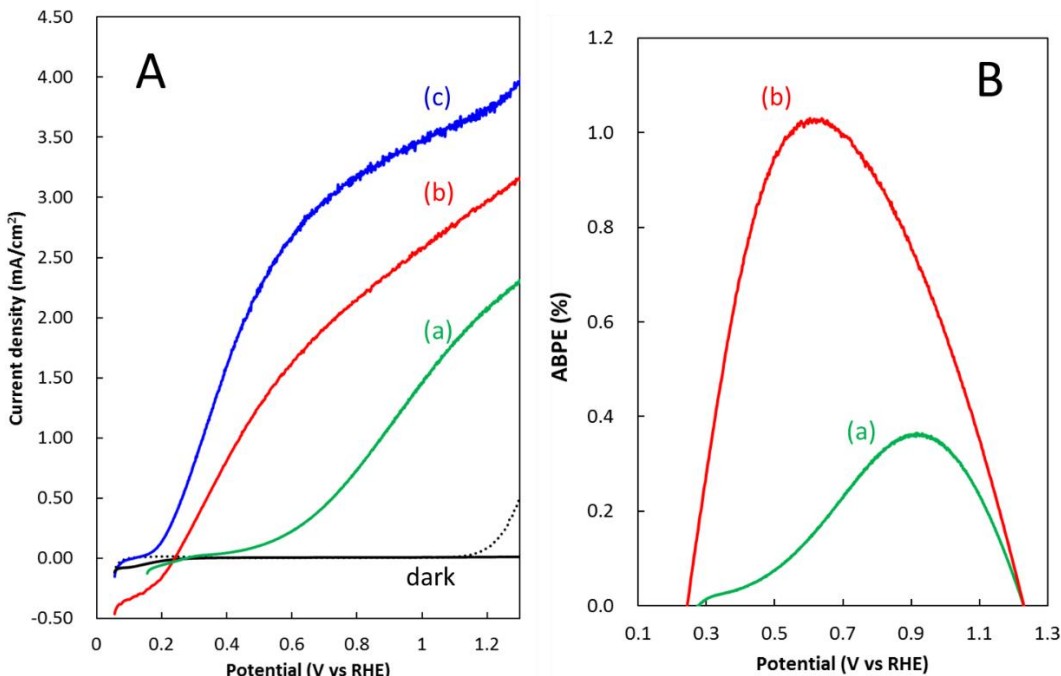

**Figure 11.** (**A**) J–V curves of: (**a**) bare $BiVO_4$; (**b**) $BiVO_4/Co_3O_4$ measured in 0.5 mol/L potassium borate (pH 9.5); and (**c**) $BiVO_4/Co_3O_4$ measured in the electrolyte with 0.2 mol/L $K_2SO_3$. The dark current curve measured without $K_2SO_3$ is shown as a black solid line and that with 0.2 mol/L $K_2SO_3$ is shown as a dotted line. (**B**) ABPE of water splitting with the photoanodes in (**A**).

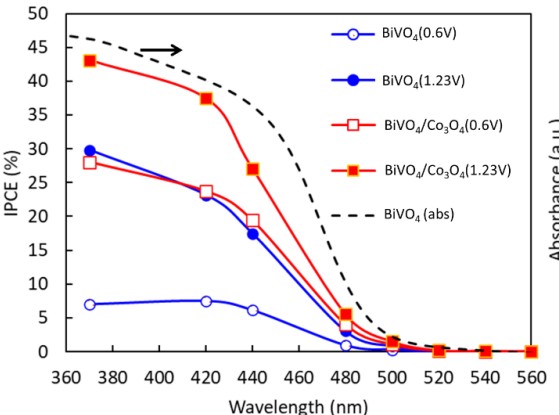

**Figure 12.** IPCE spectra for the $BiVO_4$ and the $BiVO_4/Co_3O_4$ photoanodes under front illumination at 0.6 and 1.23 V vs. RHE in 0.5 mol/L potassium borate aqueous solution (pH 9.5). The absorbance of $BiVO_4$ (black dashed line) is shown on the second vertical axis.

### 2.2.5. Impact of the Photoelectrode Area

Scale-up of photoelectrodes for the practical application of hydrogen generation from PEC water splitting is a key issue. In many reports, the photoelectrodes used were often smaller than 1 $cm^2$. These could not satisfy the actual requirements. According to previous reports [18,19], the decrease in photocurrent density by electrode scale-up is mainly due to the resistance of the FTO substrate. Here, to find the size that causes the resistance loss of the FTO substrate in this study, we probed the PEC water oxidation on the $BiVO_4$ electrodes with larger than 1 $cm^2$ from 1.3 to 3.9 $cm^2$. The current–potential (I–V) curves of the bare $BiVO_4$ photoanodes are shown in Figure 13A. Photocurrents increased with the area of the photoelectrodes. The maximum photocurrent reached 9 mA at 1.5 V vs. RHE with the 3 × 1.3 $cm^2$ photoanode. Nevertheless, the 3 × 1.3 $cm^2$ photoanode showed a little decrease in its photocurrent density (Figure 13B). The 1 × 1.3 and 1.8 × 1.3 $cm^2$ photoelectrodes were both retained at high photocurrent densities of 2.2 mA/$cm^2$ at 1.23 V vs. RHE. No photocurrent loss was observed in the FTO glass substrate with an area up to 2 $cm^2$ when using our $BiVO_4$ synthesis method.

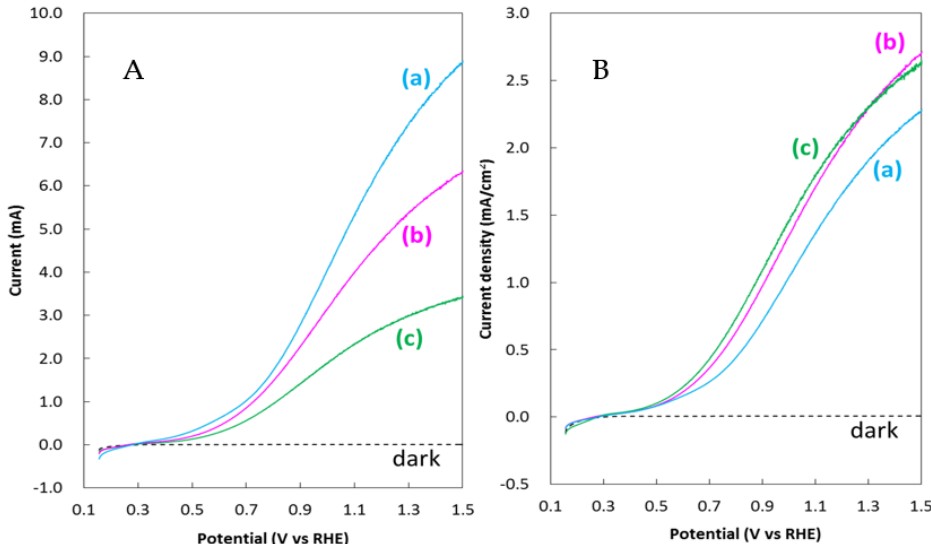

**Figure 13.** (**A**) I–V curves of the bare $BiVO_4$ film with the following sizes: (**a**) 3 × 1.3 $cm^2$; (**b**) 1.8 × 1.3 $cm^2$; and (**c**) 1 × 1.3 $cm^2$. (**B**) J–V curves calculated from the current in (**A**). The dark current is shown as a dashed line. Data were collected in 0.5 mol/L potassium borate aqueous solution (pH 9.5) under solar simulated light of 100 mW/$cm^2$ irradiation.

### 2.3. Solar-Driven Water Oxidation on the Bare $BiVO_4$ Photoanode

To confirm that the photocurrent actually causes water oxidation without any other side reaction on the bare $BiVO_4$ photoanode, PEC oxygen generation was measured by an online gas chromatography system when applying a constant potential of 1.23 V vs. RHE under solar simulated light of 100 mW/cm² irradiation. Six hours after the reaction, the stable photocurrent density dropped slightly from about 2 to 1.7 mA/cm² (Figure 14A). The corresponding oxygen generated was more than 100 μmol/cm² in 6 h, agreeing with the calculated value of the photocurrent density (Figure 14B), and its Faradaic efficiency was almost 100%. Thus, it was confirmed that the photocurrent was consistent with the solar-driven water oxidation on the bare $BiVO_4$ photoanode.

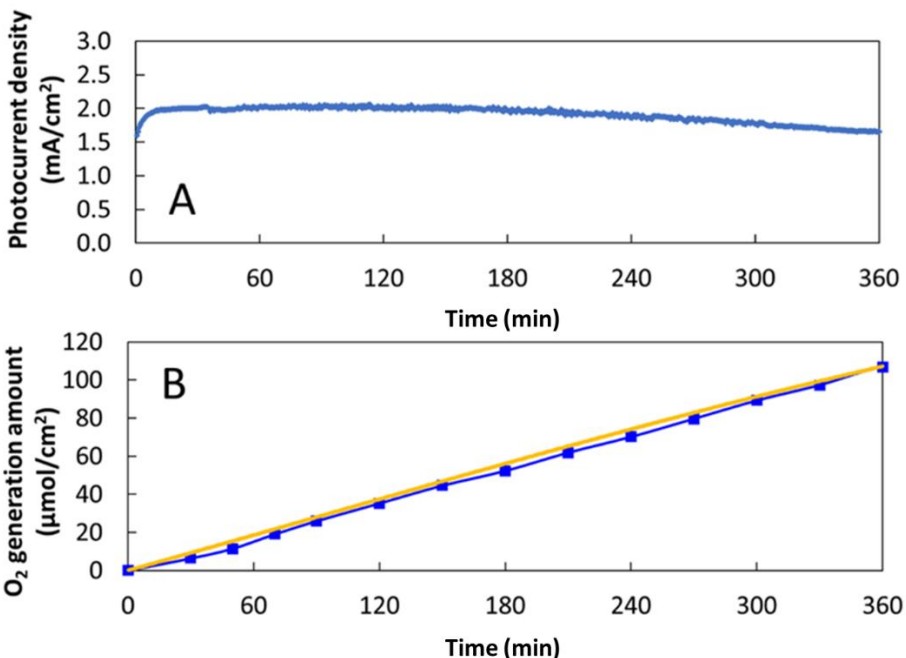

**Figure 14.** (**A**) Solar-driven water oxidation of the bare $BiVO_4$ photoanode. Data were collected at 1.23 V vs. RHE in 0.5 mol/L potassium borate aqueous solution under solar simulated light of 100 mW/cm² irradiation. (**B**) Oxygen generation amount of the bare $BiVO_4$ photoanode in (**A**). The actual value from the gas chromatograph is represented by the blue squares, and the calculated value from the photocurrent is represented by the orange line.

### 2.4. Overall Water Splitting on the $Ru/SrTiO_3$:$Rh$–$BiVO_4$ Sheet

Our $BiVO_4$ is a promising candidate for unassisted water splitting on p-n semiconductor photocatalyst sheet because of its high PEC photocurrent and transmittance. Here, as an example of the application of the present $BiVO_4$ synthesis method, visible-light-driven water splitting was performed with a $Ru/SrTiO_3$:$Rh$–$BiVO_4$ sheet. The details of the preparation of the photocatalyst sheet are presented in the experimental Section 3. Materials and Methods.

The XRD patterns of the prepared Ru (0.7 wt%)/$SrTiO_3$:Rh (1%)–$BiVO_4$ sheet was measured and matched with the reference data [37] (Figure S8). The morphological observations indicated that the electrochemically deposited $BiVO_4$ particles adhered well on the surface of the $Ru/SrTiO_3$:$Rh$ particles to form the $Ru/SrTiO_3$:$Rh$–$BiVO_4$ sheet (Figure 15). The weight of $Ru/SrTiO_3$:$Rh$ plus $BiVO_4$ was an average of 3 mg/cm² and the sheet area was 12 cm². Figure 16 shows a time course of water splitting using the $Ru/SrTiO_3$:$Rh$–$BiVO_4$ sheet. Evidently, the visible-light-driven water splitting occurred on the $Ru/SrTiO_3$:$Rh$–$BiVO_4$ sheet in deionised water without a redox mediator. It was confirmed that the stoichiometric ratio of hydrogen and oxygen was 2:1. After 22 h of water splitting, there was no change in the appearance before and after the reaction.

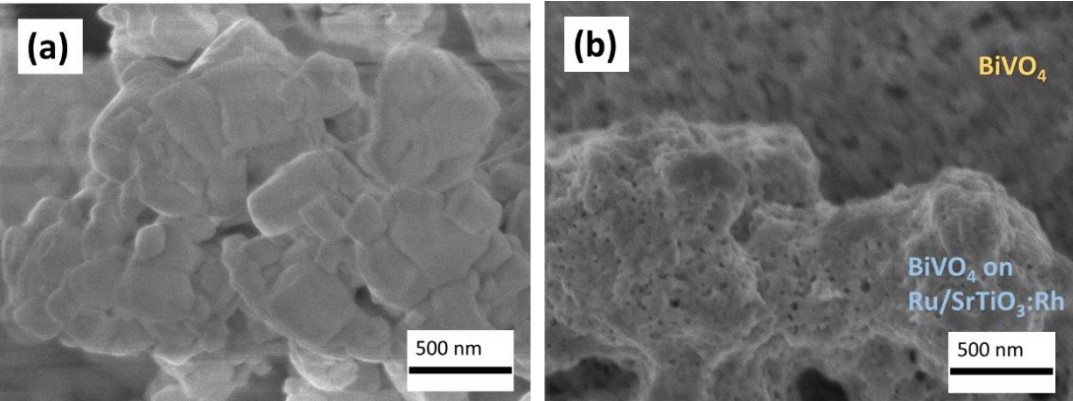

**Figure 15.** SEM images of: (**a**) the Ru/SrTiO$_3$:Rh powder; and (**b**) top-view of the Ru/SrTiO$_3$:Rh–BiVO$_4$ sheet.

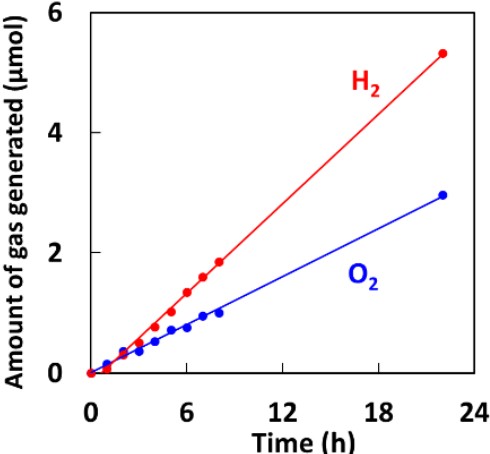

**Figure 16.** Visible-light-driven overall water splitting on the Ru/SrTiO$_3$:Rh–BiVO$_4$ sheet (area = 12 cm$^2$) in deionised water (120 mL). The substrate was an FTO glass. Light source: 300 W Xe lamp with a long-pass filter ($\lambda$ > 420 nm).

## 3. Materials and Methods

Film growth. The nanoscale porous BiVO$_4$ photoanodes (area = 1.0 × 1.3 cm$^2$) were fabricated on a conducting fluorine-doped tin oxide (FTO) coated glass (FTO-P003; <15 ohm/sq; Kaivo; Zhuhai, Guangdong, China) by a two-step electrochemical deposition at room temperature (24 ± 2 °C). A potentiostat (Hokuto Denko, HZ7000, Tokyo, Japan) with a two-electrode configuration cell was utilised for the electrochemical deposition. A 2 cm$^2$ platinum gauze was used as the counter electrode.

In Step 1, the electrodeposition bath was an aqueous solution (30 mL) containing 0.0075 mol/L bismuth (III) nitrate pentahydrate (Bi(NO$_3$)$_3$·5H$_2$O, 99.5%, Kanto Chemical Co., INC., Tokyo, Japan), 0.4 mol/L potassium iodide (KI, 99.5%, Kanto Chemical Co., INC., Tokyo, Japan), and 60 mmol/L lactic acid (C$_3$H$_6$O$_3$, 90%, Acros Organics, Fisher Scientific worldwide company, Geel, Belgium) with adjusted pH to 1.2 by nitric acid (HNO$_3$, 98.0%, Kanto Chemical Co., INC., Tokyo, Japan). Next, 20 mL of ethanol containing 0.15 mol/L p-benzoquinone (97%, Kanto Chemical Co., INC., Tokyo, Japan) were slowly added into the Bi$^{3+}$ species solution. The mixed solution was stirred overnight until a stable pH of 3.6–4.0. The BiOI was deposited at a constant bias of −0.3 V vs. Pt for 15 min. The as-deposited BiOI film was rinsed by deionised water and air-dried. To obtain the crystalline Bi$_5$O$_7$I films, the BiOI films were annealed in air for 1 h at 450 °C (ramping rate = 5 °C/min).

The same setup (as in Step 1) was employed for Step 2 deposition. To deposit the amorphous V$_2$O$_5$ on the Bi$_5$O$_7$I film, an ethanol/water solution (ratio 1:1) of 0.22 mol/L vanadyl sulphate hydrate (VOSO$_4$·nH$_2$O; >55%; Kanto Chemical Co., INC., Tokyo, Japan)

was utilised as the electrodeposition bath. The formation of amorphous $V_2O_5$ can be achieved on the $Bi_5O_7I$ film or an FTO glass by anodic deposition, e.g., at a current of 0.5–2 mA/cm$^2$ and a certain time. In this study, $V_2O_5$ was deposited on the $Bi_5O_7I$ film at a constant current of 1 mA/cm$^2$ for 15 min. The as-deposited films were annealed in air at varying temperatures at 350, 400, 450, 475, 500, and 550 °C (ramping rate = 5 °C/min) for 1 h. In the annealing process, $Bi_5O_7I$ and the amorphous $V_2O_5$ reacted to form crystallised $BiVO_4$ and $V_2O_5$. To remove the residual $V_2O_5$ on $BiVO_4$, the $BiVO_4$ photoelectrodes were immersed in a 0.5 mol/L KOH aqueous solution for 30 min with gentle stirring. Following this, these films were washed with deionised water and wiped with tissue paper (KimWipes s-200, Kimberly Clark Corp., Irving, Texas, USA) and then air dried.

The cobalt oxide cocatalyst was placed on the $BiVO_4$ films by drop-casting 3 μL/cm$^2$ of 80 mmol/L $Co(NO_3)_2·6H_2O$ (Wako Pure Chemical; 99.5%, Osaka, Japan) ethanol solution, which was followed by calcination for 1 h at 350 °C (ramping rate = 5 °C/min) in air.

Film characterisation. The morphologies of the $BiVO_4$ films were characterised by field-emission SEM JEOL JSM-6700F microscope (JEOL, Tokyo, Japan) and FIB-SEM (Zeiss Crossbeam 540, Carl Zeiss, Oberkochen, Germany). The microscope was equipped with an EDS unit (Oxford Instruments, silicon Drift Detector-X-MaxN, Abingdon, Oxfordshire, UK). The phase identification was achieved by XRD utilising a Rigaku (MiniFlex 600, Rigaku, Tokyo, Japan) X-ray diffractometer equipped with Cu Kα (40 kV and 15 mA; λ = 1.540619 Å) radiation and operated in 2θ scan modes from 10° to 65°, 0.02° step-size, 5° per min. The adopted result was the average value after five measurements. To confirm the phases of BiOI, $Bi_5O_7I$ and $BiVO_4$, reference JCPDS files for BiOI (JCPDS No. 00-010-0445), $Bi_5O_7I$ (JCPDS No. 01-071-3024), and $BiVO_4$ (JCPDS No. 00-044-0081) were cited. The ultraviolet–visible absorption spectra of the photoanodes were analysed on a spectrophotometer (JASCO V-780, JASCO, Tokyo, Japan) with an integrating sphere that was operated in the transmission mode to minimise a negative effect by light scattering.

PEC measurements. The PEC properties of the $BiVO_4$ films were measured under AM1.5G type solar simulated light (Asahi spectra, HAL320, Tokyo, Japan) with a spotlight area of 3 × 3 cm$^2$. The light output was calibrated to 100 mW/cm$^2$ by a power meter (Asahi spectra, CS-20, Tokyo, Japan) The PEC performance of the photoanodes was evaluated in a three-electrode configuration utilising a potentiostat (Hokuto Denko, HSV110, Tokyo, Japan). The reference electrode was Ag/AgCl in saturated KCl, and a 2 cm$^2$ Pt gauze was utilised as the counter electrode. As a standard, the films were illuminated from the $BiVO_4$ side (front). The scan rate for linear sweep voltammograms was set to 20 mV/s. The electrolyte comprised an aqueous solution of 0.5 mol/L boric acid that was pH-adjusted to 9.5 with potassium hydroxide. All the potentials in this work were reported against the RHE calculated from those relative to the Ag/AgCl reference electrode by Nernst equation, $E_{RHE} = E_{Ag/AgCl\,vs.\,NHE} + E_{Ag/AgCl} + 0.059\,pH$. Here, $E_{RHE}$ is the converted potential vs. RHE. $E_{Ag/AgCl\,vs.\,NHE}$ is the correction coefficient for Ag/AgCl reference electrode regarding normal hydrogen electrode (NHE) and was taken as 0.197 at 25 °C. $E_{Ag/AgCl}$ is the experimentally measured potential against the Ag/AgCl reference.

Further, ABPE (%) was calculated using Equation (10):

$$ABPE = \frac{\Delta J \times (1.23 - V)}{P} \times 100 \qquad (10)$$

where $\Delta J$ ($J_{light} - J_{dark}$ (mA/cm$^2$)) is the measured photocurrent density, V (vs. RHE) is the applied bias, and P is the incident light intensity (100 mW/cm$^2$).

The incident photon-to-current conversion efficiency (IPCE) measurements were carried out with coloured filters from 370 to 560 nm by measuring the photocurrent produced under chopped monochromatic light irradiation. The monochromatic light

intensities were measured by a photodiode power sensor (OPHIR Japan Ltd., Tokyo, Japan). The IPCE (%) was then calculated using Equation (11):

$$\text{IPCE}(\lambda) = \frac{1240j(\lambda)}{I_o(\lambda) \times \lambda} \times 100 \tag{11}$$

where $\lambda$ is the wavelength of incident monochromatic light (nm), $j(\lambda)$ is the difference between dark current density and photocurrent density under illumination at wavelength $\lambda$ (mA/cm$^2$), and $I_0(\lambda)$ is the incident light intensity at wavelength $\lambda$ (mW/cm$^2$).

Measurement of the oxygen produced by PEC water splitting. PEC water splitting on the BiVO$_4$ film was performed at an applied potential of 1.23 V vs. RHE under solar simulated light (Asahi spectra, HAL320, Tokyo, Japan) at 100 mW/cm$^2$ irradiation (Asahi spectra, CS-20). The PEC properties were evaluated using a potentiostat (Hokuto Denko, HZ-7000, Tokyo, Japan). An airtight three-electrode configuration with a compartmental H-type cell that was separated by a membrane (DuPont, Nafion 117, Wilmington, Delaware, USA) was employed. Incidentally, using a membrane separation cell could prevent the occurrences of reverse reactions. The reference electrode was Ag/AgCl in saturated KCl, and a Pt wire (diameter = 0.5 cm, length = 6 cm) was utilised as the counter electrode. Furthermore, 0.5 mol/L potassium borate (buffered at pH 7.5) was utilised as an electrolyte. The amount of evolved oxygen on the BiVO$_4$ photoanode was recorded by an online gas chromatograph (GC) (Shimadzu GC-8A, Kyoto, Japan), TCD detector, MS-5 Å column, oven at 70 °C, argon carrier). Argon was the carrier gas from the reactor to the GC instrument at a flow rate of 15 mL/min.

Preparation and water splitting of the photocatalyst sheet. Rh-doped SrTiO$_3$ powder was prepared by a previously reported solid-state method [37]. In a typical procedure, SrCO$_3$ (Kanto Chemical, 99.9%, Tokyo, Japan), TiO$_2$ (High Purity Chemicals, 99.9%, Tokyo, Japan), and Rh$_2$O$_3$ (Wako Pure Chemical, Tokyo, Japan) were mixed in the Sr/Ti/Rh ratio of 1.07:0.93:0.01. The mixture was calcined in air for 1 h at 900 °C, after which the powder was mingled well and calcined again for 10 h at 1100 °C. Ru was utilised as a cocatalyst to SrTiO$_3$ for hydrogen evolution. Ru (0.7 wt%) was loaded on SrTiO$_3$:Rh by photo-assisted reduction in an aqueous 10 vol% methanol solution containing RuCl$_3$ (Wako Pure Chemical, 85.0%). The Ru/SrTiO$_3$:Rh photocatalyst was collected by filtration and washed with distilled water. The photocatalyst powder (100 mg) was added to 0.3 mL of a mixture of 1:2 acetylacetone (Kanto Chemical, 99.5%) and deionised water, and then sonicated for 5 min. The resulting paste was placed on an FTO glass substrate. Next, to remove the organic solvent, the photocatalyst sheet was heated in air for 2 h at 200 °C. By the method described above (Figure 1), BiVO$_4$ was deposited on the Ru/SrTiO$_3$:Rh sheet. The overall water-splitting experiment on the Ru/SrTiO$_3$:Rh–BiVO$_4$ sheet was performed in 120 mL of deionised water without a redox mediator. A top-irradiation cell with a Pyrex window was utilised after degassing by applying a vacuum and purging with argon. The setup was returned to reduced pressure (70 Torr) and irradiated with a 300 W Xe arc-light source (PerkinElmer; CERMAX PE300BF, Waltham, Massachusetts, USA) with a long-pass filter ($\lambda > 420$ nm). The amounts of evolved hydrogen and oxygen were determined by an online GC (Shimadzu; GC-8A, TCD, MS-5Å column, oven at 50 °C, argon carrier).

## 4. Conclusions

We developed a highly efficient BiVO$_4$ photoanode synthesis method for PEC water oxidation by a simple electrochemical procedure. According to our method, the electrochemical deposited amorphous V$_2$O$_5$ was able to spread densely among the precursor Bi$_5$O$_7$I flakes. Therefore, during the appropriate annealing process of Bi–V–O conversion at 475 °C for 1 h, the BiVO$_4$ was formed along the shape of the Bi$_5$O$_7$I flakes by converting with the amorphous V$_2$O$_5$, thereby creating a porous structured film. Since the Bi–V–O reaction occurred throughout the thin film as the same time, the BiVO$_4$ particles were highly monodispersed nanoscales.

It is considered that the BiVO$_4$ formation control by the addition of lactic acid strengthened the atomic arrangement of the (010) orientation and improved the transmittance of the BiVO$_4$ photoanode thin film. The (010) facets were the active sites for photoelectron migration to the FTO substrate, which enhanced charge separation and reduced the electron$-$hole pair recombination. Thus, high PEC properties were accomplished even when the light irradiation was from the front of the BiVO$_4$ photoanode.

The optimised BiVO$_4$ photoanode in this study showed a photocurrent density of 2.2 mA/cm$^2$ at 1.23 V vs. RHE under one sun light illumination. The maximum ABPE was 0.37% at 0.9 V vs. RHE. The IPCE at 420 nm achieved 23.2% at 1.23 V vs. RHE. Both the enhancement in ABPE and IPCE were higher than the values reported recently. After 6 h of reaction the PEC water oxidation appeared to have excellent stability. The Faradaic efficiency of ca. 100% and high oxygen evolution of over 100 μmol/cm$^2$ were obtained. Furthermore, increasing the BiVO$_4$ photoanode area to 2 cm$^2$ did not reduce the photocurrent density. When the area reached 3 cm$^2$, the photocurrent densities decreased a little. The BiVO$_4$ photoanode has high light transmittance and will be suitable for practical use of PEC tandem cell system. It was also found that even higher efficiency can be obtained from the effect of surface modification of the Co$_3$O$_4$ cocatalyst. The photocurrent density of the BiVO$_4$/Co$_3$O$_4$ photoelectrode was 2.9 mA/cm$^2$ at 1.23 V vs. RHE under one sun light illumination. The maximum APBE was 1.03% at 0.6 V vs. RHE, and IPCE at 420 nm carried out at 1.23 V vs. RHE was 37.5%.

In addition, it was confirmed that the BiVO$_4$ formed by our new two-step method could attach firmly to a Ru/SrTiO$_3$:Rh photocatalyst sheet. The Ru/SrTiO$_3$:Rh–BiVO$_4$ photocatalyst sheet successfully performed visible-light-driven overall water splitting in deionised water for 22 h.

**Supplementary Materials:** The following are available online at https://www.mdpi.com/2073-4344/11/1/136/s1, Figure S1: ultraviolet–visible absorption spectra, Figure S2: photographs show hydrophobicity, Figure S3: EDS elemental analysis spectrum, Figure S4: enlarged view of XRD patterns, Figure S5: Top view SEM images, Figure S6: ultraviolet–visible absorption spectra (Temp.), Figure S7: Open circuit potential (Temp.), Figure S8: XRD patterns of Ru/SrTiO$_3$:Rh–BiVO$_4$ sheet.

**Author Contributions:** S.H.-K. designed the experiments and wrote the manuscript. A.K. supervised this study. W.S. and Y.Y. supported the experiments and contributed to the scientific discussion. All authors have read and agreed to the published version of the manuscript.

**Funding:** S. H.-K was funded by FDCT for this study, grant number: FDCT 008/2017/AMJ and the APC was funded by Photocatalysis Center of TUS.

**Institutional Review Board Statement:** Not applicable.

**Informed Consent Statement:** Not applicable.

**Data Availability Statement:** Data is contained within the article.

**Acknowledgments:** S.H.-K. is grateful to the Science and Technology Development Fund from Macau SAR (FDCT 008/2017/AMJ) and the Multi Year Research Grant from the University of Macau (MYRG2019-00005-IAPME) for funding.

**Conflicts of Interest:** The authors declare no conflict of interest.

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
