# Peer review of "Preparation of Nanoparticle Porous-Structured BiVO4 Photoanodes by a New Two-Step Electrochemical Deposition Method for Water Splitting"

_catalysts, doi:10.3390/catal11010136_

Round 1

Reviewer 1 Report

The study of new oxide semiconductors for water splitting ability under visible light and development of facile their synthesis is an urgent problem. BiVO4 is an n-type semiconductor that has recently emerged as one of the most promising photoanodes for use in water-splitting. Therefore, the article can be published in the Catalysts. But for me, some unclear points remain.
1. BiVO4 has been prepared by the thermal conversion of
electrodeposited BiOI and was demonstrated by McDonald and
Choi in (McDonald, K. J.; Choi, K.-S. A new Electrochemical Synthesis Route for a BiOI Electrode and Its Conversion to a Highly Efficient Porous BiVO4 Photoanode for Solar Water Oxidation. Energy Environ. Sci. 2012, 5, 8553−8557.), but this is not cited in the manuscript. Is the proposed method differs from this?
2. Does the formation of new phases occur during thermal calcination of Co(NO3)2 ethanol solution for 1 hour at 350°C in the air?
3. J-V curves of BiVO4 were measured in different electrolytes: K2SO3 (Fig. 10 and 12) and potassium borate aqueous solution (Fig. 8 and 9) which complicates their comparison.
4. It needs to calculate the incident photon-to-current conversion efficiency for water decomposition under visible light irradiation BiVO4 thin film on a conducting glass electrode.
5. What is the stability of the BiVO4 electrode?

Reviewer 2 Report

In this paper, the authors presented a new electrochemical deposition method for the preparation of nanoparticle porous-structured BiVO4 photoanodes. The proposed procedure allowed the even distribution of V2O5 amongst the BiOI to proceed the Bi–V–O reaction uniformly.

The paper is properly divided in sections and sub-sections, but it needs some corrections before its publication on the journal.

  • The authors should check the language since some errors are present in the text. For example, at line 42 the statement seems to be incomplete;
  • Did the authors perform any adhesion test in order to investigate the strength of the contact of the deposited BiVO4 particles on the surface of the Ru/SrTiO3∶Rh particles to form the Ru/SrTiO3:Rh–BiVO4 sheet?
  • The authors should move the conclusions section after the material and methods section;
  • Did the authors observe any modification of the Ru/SrTiO3:Rh–BiVO4 sheet after the water splitting reaction?
  • Did the authors perform any longer tests in order to assess the stability of the Ru/SrTiO3:Rh–BiVO4 sheet?
  • Why did not the authors report the curve relevant to sample C in figure 11B?
  • The authors should compare the performance of their catalyst with the ones of some other catalysts in literature;
  • The authors should extend the literature survey, by including more recent papers in the field. Some examples are: 10.1016/j.nanoms.2019.10.003, 10.1021/acsami.8b11241, 10.1021/acsomega.0c00699.

Reviewer 3 Report

Accept as it is

Round 2

Reviewer 2 Report

The authors well improved the manuscript. In my opinion, it can be published on the journal.